# Predicting the risk of asthma attacks in children, adolescents and adults: protocol for a machine learning algorithm derived from a primary care-based retrospective cohort

Zain Hussain,[1] Syed Ahmar Shah [ID],[1,2] Mome Mukherjee [ID],[1,2] Aziz Sheikh[1,2,3]

[1]Usher Institute, Edinburgh Medical School, The University of Edinburgh, Edinburgh, UK
[2]Asthma UK Centre for Applied Research (AUKCAR), The University of Edinburgh, Edinburgh, UK
[3]Division of Community Health Sciences, The University of Edinburgh, Edinburgh, UK

**Correspondence to**
Dr Syed Ahmar Shah;
ahmar.shah@ed.ac.uk

## ABSTRACT

**Introduction** Most asthma attacks and subsequent deaths are potentially preventable. We aim to develop a prognostic tool for identifying patients at high risk of asthma attacks in primary care by leveraging advances in machine learning.

**Methods and analysis** Current prognostic tools use logistic regression to develop a risk scoring model for asthma attacks. We propose to build on this by systematically applying various well-known machine learning techniques to a large longitudinal deidentified primary care database, the Optimum Patient Care Research Database, and comparatively evaluate their performance with the existing logistic regression model and against each other. Machine learning algorithms vary in their predictive abilities based on the dataset and the approach to analysis employed. We will undertake feature selection, classification (both one-class and two-class classifiers) and performance evaluation. Patients who have had actively treated clinician-diagnosed asthma, aged 8–80 years and with 3 years of continuous data, from 2016 to 2018, will be selected. Risk factors will be obtained from the first year, while the next 2 years will form the outcome period, in which the primary endpoint will be the occurrence of an asthma attack.

**Ethics and dissemination** We have obtained approval from OPCRD's Anonymous Data Ethics Protocols and Transparency (ADEPT) Committee. We will seek ethics approval from The University of Edinburgh's Research Ethics Group (UREG). We aim to present our findings at scientific conferences and in peer-reviewed journals.

## Strengths and limitations of this study

► Comparison of a variety of machine learning approaches with a logistic regression model to develop a prognostic tool for predicting asthma attacks will be done.
► First study to apply novelty detection (a one-class classifier) for predicting asthma attacks using primary care data.
► Standardised performance evaluation measures will be used when comparing machine learning algorithms.
► A very large national primary care dataset will be utilised, with a population of people with asthma, which will increase the likelihood that results will be generalisable.
► Some potentially important risk factors, such as inhaler technique and allergen exposure are absent from this database, and we will not be validating this work against another dataset.

million primary care consultations, 93 000 hospital inpatients episodes and 1400 deaths have been attributed to asthma, costing the UK public sector over £1.1 billion.[2]

Asthma is a variable condition. Most people with asthma thus have long periods where they are either asymptomatic or experience only relatively mild symptoms and then experience asthma attacks, which may prove life-threatening. Asthma self-management plans aim to support patients/carers in identifying when asthma control is deteriorating and then encouraging patients to modify treatment accordingly to improve asthma control. This is at present largely a qualitative process as there is no widely used algorithm to help predict the risk of an asthma attack.

The limited existing body of evidence on this subject reveals that most investigators have employed univariate regression modelling to identify one or more risk factors for

## INTRODUCTION

Asthma is a common and heterogeneous disease that affects approximately 300 million people worldwide; in most parts of the world its prevalence is growing or remaining stable at best.[1] Around 180 000 deaths are attributed to asthma per year, the majority of which are preventable. Asthma attacks (also termed asthma exacerbations) can affect people with asthma of any age, ethnicity and severity. In the UK, there are each year at least 6.3

BMJ

asthma attacks. Such studies assess the independent contribution of each of several factors, and do not determine the predictive performance of an optimal combination of factors in individual patients. In contrast, a limited number of studies have attempted to combine various risk factors in order to develop a risk scoring algorithm. A comprehensive systematic review by Loymans et al[3] aimed to identify and critically appraise predictive models which assess future risk of asthma attacks. They were able to identify 24 models from 12 studies, using a search up to April 2017, in which multiple factors (referred to as predictors henceforth) were accounted for and predictive performances were evaluated. They concluded that a generalisable model for predicting severe attacks is feasible, as the predictive properties of most models were comparable in two distinct validation populations, and that there is scope for improved performances. A major limitation of the studies reviewed was the small population size. Except one study,[4] all studies reviewed had a population size of less than 8000, and consequently a low number of asthma attacks. Variations in model performance was attributed to design methodologies, reporting and differences in asthma outcomes. The definition of an asthma attack varied between studies, with some using just one, or combination, of the following: prescription of oral corticosteroids, attending the accident and emergency department and/or hospital admission for asthma. The definition of uncontrolled asthma also varied, with some studies using subjective measures, such as data in patient diaries. We have identified Blakey et al[5] as our benchmark study (excluded from Loymans et al for reporting relative risk rather than absolute risk), as it successfully addresses many of our aforementioned limitations associated with developing a risk score. This study effectively identified patients at risk of recurrent attacks using the Optimum Patient Care Research Database (OPCRD) database, which contained longitudinal medical records of 118 819 patients with asthma, and was broadly representative of the general asthma population in the UK. Various predictors (or features) for attacks were identified and evaluated using logistic regression—building on the work by Price et al[6]—and entered into a multivariate logistic regression analysis with feature selection using backward elimination, based on the importance of individual features. The resultant risk score for recurrent attacks over a 2-year outcome period, was used to develop an online asthma risk prediction tool for research and clinical purposes.[7] There is however a need to build on this study, as it only utilised logistic regression, which has been shown to be a poor classifier in cases of class imbalance (ie, disproportionate ratio of event vs no event data) which is the case for asthma attacks in a population.[8] In addition, logistic regression is a two-class classifier (ie, it requires reasonable number of samples of both classes to adequately model the data). Two class-classifiers are the most common approach and all previous studies have attempted to model asthma attack using a two-class classifier. However, in situations where an event is rare (such as asthma attacks in this study), novelty detection (one-class classifier) can perform better as it only requires examples of one class that is common (in our case, it would be the periods over which an individual had no asthma attacks) to build a model. While novelty detection has been applied in several healthcare applications such as detection of masses in mammograms,[9] condition monitoring of patients in intensive care unit[10] and identification of children with infection,[11] we are not aware of any previous study that has applied novelty detection for the prediction of asthma attacks.

Since machine learning algorithms vary in their predictive abilities dependent on the underlying application and distribution of data, a range of methodologies need to be employed and compared in an attempt to improve performance.

### Research aims

We propose leveraging advancements in machine learning by systematically evaluating different modelling approaches, to develop a prognostic tool for asthma attacks that is an improvement over the current state-of-the-art logistic regression-based approach.[6]

Specifically, we aim to:

1. Identify significant risk factors associated with asthma attacks in children, adolescents and adults (aged 8–80 years), and appropriately select these for inclusion in our analysis.
2. Systematically apply several machine learning algorithms (both one-class classifier and two-class classifiers) to predict the risk of asthma attacks, over 3-month, 6-month, 12-month and 24-month outcome periods.
3. Comparatively evaluate performances of these predictive models with each other and against the benchmark logistic regression model, to identify which is the most accurate.

## METHODS

Figure 1 provides an overview of our methodology. Once the data have been extracted, it will be divided into training and testing sets, using k-fold cross-validation. The training data will then be used to develop a risk prediction model. The key steps for developing this model are feature selection and training a classifier model. The trained classifier will then be tested on the remaining data (the testing set) for validation. It is important to point out that some methods combine feature selection and classifier within a single step known as embedded methods (hence the dashed line combining feature selection with classifier).

### Patient and public involvement

We aim to work closely with the members of the Patient and Public Involvement group of the Asthma UK Centre for Applied Research, University of Edinburgh during this research. We will seek input from the PPI group to comment on the findings of the study and help us in disseminating the key findings to the public via social media, website and various public engagement activities.

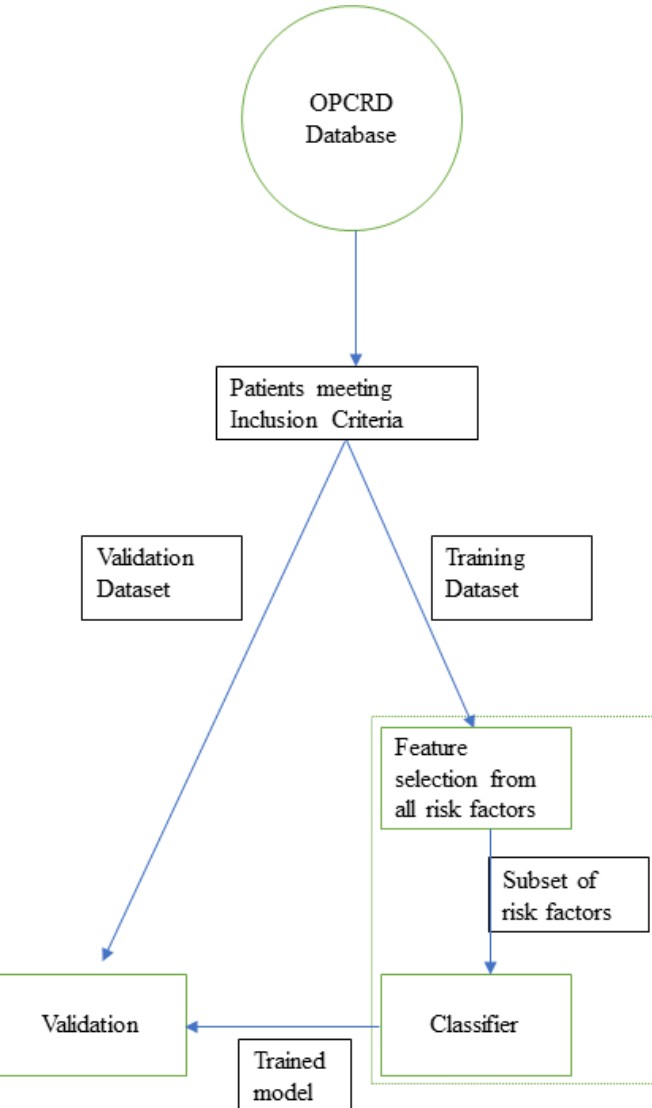

**Figure 1** Flowchart of proposed steps in the methodology. OPCRD,Optimum Patient Care Research Database.

## Study design and population

We will conduct a retrospective cohort study, using OPCRD, a longitudinal deidentified primary care database of over 6.3 million patients from over 700 general practices in the UK. The medical record data for each patient will include demographic information, disease diagnoses in the form of Read codes, drug prescriptions, medical test results and hospitalisation information. The three most recent years of continuous data for each patient will be analysed, from which 1 year will be for baseline characterisation, and the other two will form the outcome data.

The study population will consist of patients with actively treated asthma ('asthma diagnostic' Read codes prior to study commencing, and a current asthma prescription), aged 8–80 years and with three or more years of continuous data. This study will focus on adults and young people aged 8 years and over. Missing data on age and/ or sex, will result in patient exclusion; this along with any other patient exclusions will be documented. We will not attempt to exclude patients with comorbidities. We have, however, included comorbidities as one of the candidate predictors (see table 1) that will allow us to adjust for any potential confounders arising from comorbidities.

Active asthma will be defined by a prescription with ≥ 2 asthma drugs during year 1 of the study, to include any of short-acting beta agonists, long-acting beta antagonists (LABA), inhaled corticosteroids (ICS), fixed-dose ICS/ LABA combination, leukotriene receptor antagonists and/or theophylline, along with the absence of a Read code for resolved asthma at any point during the 3-year study period. For all characteristics derived from Read codes, full code lists will be provided as online supplementary materials.

## Potential risk factors

The candidate predictors for asthma attacks, to be assessed for inclusion in the models will be from the baseline study by Blakey et al,[5] which selected measures that are routinely collected in primary care, as shown in table 1. Variables with missing data, other than age and/or sex (exclusion criteria), will be dealt with using multiple imputation.[12]

## Outcome ascertainment

The primary outcome for each model will be the occurrence of an asthma attack, as defined by the European Respiratory Society/American Thoracic Society,[13] namely prescription of oral corticosteroids following an acute presentation of asthma, an accident and emergency attendance for asthma or an asthma-related hospitalisation.

## Analysis plan
### Feature selection

An important stage, prior to the application of many well-known classifiers, is the selection of appropriate risk factors, termed 'feature selection' in machine learning. This works to avoid the problem of overfitting, thereby increasing generalisability, and improving model accuracy. We will first generate a correlation heat map to visually assess which features are most correlated with an asthma attack.[14] Univariate analysis, such as correlation heatmaps, do not take the interaction of features into consideration. We will, therefore, subsequently use the ReliefF algorithm that will allow us to rank features based on their predictive power.[15]

### Classification

Supervised classification algorithms will be used to obtain a classifier which can differentiate between stable asthma and a patient profile that is at risk of an attack. Figure 2 provides an overview of the classification algorithms that we will explore. Broadly, there are two types of supervised classification algorithms that we will explore: one-class classifier and a two-class classifier.

The first step within a one-class classifier is to learn a probabilistic model of normality in $n$ dimensions (where $n$ is the number of features). There are two main types of models for learning the unconditional probability density function characterising normality: Parzen Windows and

**Table 1** Candidate predictors to be assessed for inclusion in models (adapted from Blakey et al[5])

| Variable | Description |
|---|---|
| Sex | Male or female |
| Age | In years at the start of the 3-year study period |
| BMI | Last recorded, in kg/m$^2$; categorised as underweight (<18.5), normal (18.5–24.9), overweight (25-29.9) or obese (≥30) |
| Ethnicity | Ethnicity information if available (white, black, Asian, South Asian Caribbean etc) |
| Smoking status | Last recorded, categorised as never smoker, current smoker or ex-smoker |
| Charlson comorbidity index | Score in the baseline year, categorised as 0, 1–4, 5–9, ≥10 |
| Comorbidities* | Recorded ever or active: eczema, allergic and non-allergic rhinitis, nasal polyps, anaphylaxis diagnosis, anxiety/depression diagnosis, diabetes (type 1 or 2), GERD, cardiovascular disease, ischaemic heart disease, heart failure, psoriasis |
| Comedications | In baseline year, prescription (yes/no) for paracetamol, NSAIDs, beta-blockers, statins |
| % predicted PEF | Recorded ever, expressed as percentage of predicted normal, categorised as unknown,<60%, 61%–79% and ≥80% |
| Blood eosinophil count | Last recorded, in 109 cell/L, categorised as ≤0.4 or >0.4 |
| BTS step† | |
| Step 1 | Inhaled SABA as needed |
| Step 2 | ICS or LTRA |
| Step 3 | Add LABA to ICS or use high-dose ICS (≥400 mg/day FP equivalent) |
| Step 4 | Add LTRA/Theo to (ICS+LABA) or add LABA/LTRA/Theo to high-dose ICS |
| Step 5 | Add OCS |
| Average daily dose of SABA/ICS | Cumulative dose of SABA/ICS prescribed in baseline year, expressed in mg/day albuterol or FP equivalent and divided by 365.25 |
| Prescribed daily ICS dose | Dose of ICS prescribed at last prescription of baseline year in mg/day, FP equivalents |
| ICS medication possession ratio | ICS refill rate during the baseline year: sum of number of days per pack (number of actuations per pack/number of actuations per day)/365.25 |
| ICS device type | In baseline year: categorised as no ICS, MDI, BAI or DPI |
| Spacer use with ICS pMDI | Recorded in baseline year (yes/no) |
| Oral corticosteroid use | Any maintenance prescription for corticosteroids in baseline year (yes/no) |
| Prior asthma education | Recorded ever (yes/no) |
| Primary care consults | Number of primary care consultations, categorised as 0, 1–5, 6–12, ≥13 |
| Primary care consults for asthma | Number of primary care consultations with an asthma-related Read code |
| Antibiotics with lower respiratory consult | Number of consultations that resulted in antibiotic prescription (included to capture asthma events that may have been misclassified as LRTI) |
| Acute respiratory events | Number of events in the baseline year, defined as asthma-related hospitalisation or ED attendance or an acute course of OCS or antibiotics prescription with lower respiratory consultation |
| Acute OCS courses | Number of acute courses of OCS in baseline year, categorised as 0, 1, ≥2 |
| Acute OCS courses with lower respiratory consult | Number of OCS courses with Read code for lower respiratory consultation in baseline year, categorised as 0, 1, ≥2 |
| Antibiotics courses | Number of antibiotics prescriptions with Read code for lower respiratory consultation in baseline year, categorised as 0, 1, ≥2 |
| Hospital attendance/admission | Number of asthma-related‡ ED, inpatient and outpatient attendance/admission in baseline year (as recorded in primary care data) |
| Asthma attacks | Number of asthma-related‡ hospital ED attendance, inpatient admission or acute OCS course |
| Eosinophil count | Blood eosinophil count (cells/L) categorised into high and not high (threshold of $0.35 \times 10^9$ cells/L) to define high/not high eosinophil count[24] |

Continued

| Table 1 | Continued |
| --- | --- |
| Variable | Description |

*Comorbidity recorded 'ever' was defined as a diagnostic Read code during the baseline year or at any time before baseline. 'Active' refers to those for which a diagnosis was recorded within the baseline year and/or a previous diagnosis was accompanied by a prescription for the comorbidity within the baseline year. 'Rhinitis' included allergic and non-allergic rhinitis.
†Based on the British guideline on the management of asthma (October 2014) for adults and children.[25]
‡Any patient with a lower respiratory Read code (asthma or LRTI code).
BAI, Breath-actuated inhaler; BMI, body mass index; BTS, British Thoracic Society; DPI, dry powder inhaler; ED, emergency department; FP, fluticasone propionate; GERD, gastro-oesophageal reflux disease; ICS, inhaled corticosteroids; LABA, long-acting beta antagonists; LRTI, lower respiratory tract infection; LTRA, leukotriene receptor antagonist; MDI, metered-dose inhaler; NSAIDs, non-steroidal anti-inflammatory drugs; OCS, oral corticosteroids; PEF, peak expiratory flow; SABA, short-acting b2 agonist; Theo, theophylline.

Gaussian Mixture Modelling.[16] Following the learning or training phase, the data fusion model of normality is used to evaluate the probability that the features acquired from a subject in a test set (ie, a subject not included in the training set) can be considered to be normal. Novelty detection identifies those subjects with features outside the distribution of normality.

Two-class classifiers can broadly be categorised into discriminative models (these methods learn a decision boundary using the training data) and generative models (these models learn the underlying probability distribution of the data and then use Bayes formula for classification). For generative models, we will use the naive Bayes algorithm. For discriminative models, we will first use logistic regression—the most commonly used method in the field of medicine, hence will be used as our benchmark approach (as was done in Blakey *et al*[5]). However, logistic regression can only learn a linear decision boundary in the feature space and we will subsequently apply additional discriminative classification algorithms in our pursuit of developing a prognostic tool. This comparative approach is used commonly in the field of machine learning.[17] The additional algorithms we will explore are 'Support Vector Machines' (SVMs) which seek to find an optimal hyperplane in n-dimensional space for separating two classes (the maximum-margin hyperplane). In SVM, the decision points are defined by only the training points that are closest to the boundaries

(the support vectors). By using a 'kernel trick', the SVM can be used to find a non-linear boundary. We will also explore the use of 'Decision Trees' which are widely used in healthcare due to their simplicity and ease of interpretability. The aforementioned algorithms were selected based on evidence in the literature on their use in predictive modelling.[18 19] Lastly, traditional feature selection techniques can be sensitive to random error, hence we also propose the use of Least Absolute Shrinkage and Selection Operator (LASSO).[20] LASSO is a modification of logistic regression where the cost function includes an additional term (a regularisation term which is the sum of the absolute values of the unknown parameters to be found). Effectively, it is an embedded technique that combines classification learning and feature selection by systematically removing (or knocking out) features based on their predictive ability. This technique can thus help us rank features based on their importance (ie, predictive power) as well as enhance the prediction accuracy and interpretability of the classifier.

### Performance evaluation

K-fold cross-validation will be used to estimate the predictive accuracy of each machine learning model on unseen data. Each round of cross-validation involves the splitting of the dataset into subsets (the parameter K refers to the number of subsets), following which the algorithm is trained on one subset (termed the 'training set'), and is

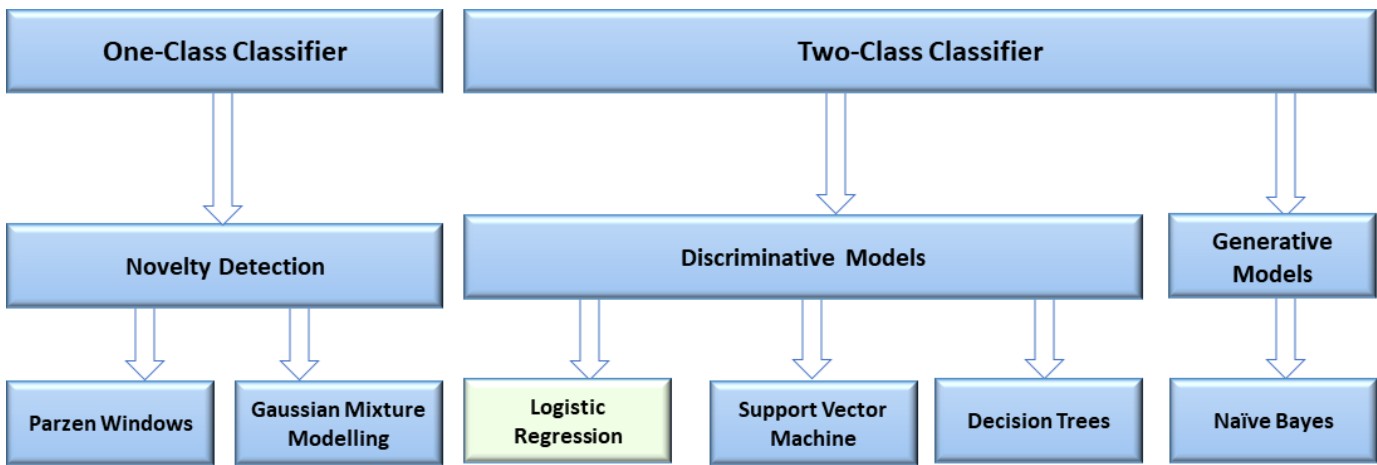

**Figure 2** Overview of the various classification algorithms that will be used to predict asthma attack. The methods are broadly divided into one-class classifier and two-class classifier. Our baseline reference method is logistic regression (shown in green).

then validated against another subset (termed the 'validation set' or 'testing set'). In order to reduce variability, multiple rounds of cross-validation are performed, with different subsets from the same dataset, and the average cross-validation error is used as a performance indicator.

Model evaluation will be carried out quantitatively via receiver operator characteristic (ROC) analysis. ROC curves will be constructed to evaluate and compare the predictive models. A ROC curve plots sensitivity (ie, the proportion of positive cases that are correctly identified) and specificity (ie, the proportion of negative cases that are correctly identified) at a range of threshold settings. The area under the curve provides an aggregate measure of model performance across all classification thresholds and will be used to compare predictive models.

## DISCUSSION

This is the first study that will leverage advances in machine learning to develop a prognostic tool for asthma attacks using a UK-wide dataset. It is also the first study that will apply a one-class classifier (novelty detection) to predict asthma attack using routinely collected primary care data. It will fill an important gap in the evidence base, as similar studies carried out to-date have utilised single models (primarily logistic regression and lacking comparative algorithm analysis) and none have utilised novelty detection. A study protocol[21] was recently published, for developing a machine learning-based prediction tool for asthma attacks, which will employ novel applications of established machine learning. However, the model will be derived and validated using data across Scotland, and the work will only focus on two-class classifiers. Our study will be based on a general population of people with asthma, with data obtained from a large UK-wide dataset. Our findings will therefore be applicable to patients undergoing treatment for asthma in the UK and can potentially inform clinical practice. A variety of machine learning algorithms have been selected for comparison, based on evidence from the literature on their uses, and the performance evaluation measures to be used are also standard, well-accepted approaches.

One limitation of our study is the absence of some potentially important risk factors from the database (based on reports from previous studies[22]), including allergen exposure and inhaler technique, along with the lack of another independent dataset for validating models. Another limitation is not all asthma attacks will be captured in routinely collected structured electronic health records. There is the potential to use natural language processing-based approaches to interrogate the free text records and this may increase the accuracy with which such events are detected. Furthermore, we do not have access to pharmacy records for prescription data (which may help us better estimate patient adherence to medication prescription) and would therefore use prescription records to determine patient usage which may not always be the correct.

Following the development of our prognostic tool, independent validation will be required using another large dataset. Prospective trials will then be needed in order to evaluate the implementation of the model in clinical practice, along with its effects on asthma-related outcomes in the population.

## ETHICS AND DISSEMINATION

All authors with data access have completed the Safe Users of Research data Environment training, provided by the Administrative Data Research Network. All analysis will be conducted in concordance with the National Services Scotland Electronic Data Research and Innovation Service (eDRIS) user agreement.

Our study protocol has been reviewed and ethically approved by The Anonymous Data Ethics Protocols and Transparency (ADEPT) committee, thereby receiving ADEPT Approval and access to the OPCRD.

The findings from this study will be reported in line with recommendations from the TRIPOD (transparent reporting of a multivariable prediction model for individual prognosis or diagnosis) and RECORD (reporting of studies conducted using observational routinely collected health data) checklists.[23] Code scripts used for all components of the data cleaning, compiling and analysis will be made available in the open source GitHub website at https://github.com/syedahmar. We aim to present the findings at national and international conferences, and publish them in leading peer-reviewed journals.

**Acknowledgements** The authors would like to thank Dominic Ng and Moiz A Shah for their contributions to the proof reading of this manuscript. We thank OPC for access to the OPCRD dataset.

**Contributors** ZH, SAS and AS conceived and planned the analysis. SAS with ZH wrote the first draft, with contributions from all authors (ZH, SAS, MM and AS). All authors approved the final version and jointly take responsibility for the decision to submit this manuscript to be considered for publication.

**Funding** Asthma UK Centre for Applied Research; The University of Edinburgh's Chancellor's Fellowship Scheme; BREATHE - The Health Data Research Hub for Respiratory Health [MC_PC_19004] funded through the UK Research and Innovation Industrial Strategy Challenge Fund and delivered through Health Data Research UK.

**Competing interests** None declared.

**Patient and public involvement** Patients and/or the public were involved in the design, or conduct, or reporting or dissemination plans of this research. Refer to the Methods section for further details.

**Patient consent for publication** Not required.

**Provenance and peer review** Not commissioned; externally peer reviewed.

**ORCID iDs**
Syed Ahmar Shah http://orcid.org/0000-0001-5672-0443
Mome Mukherjee http://orcid.org/0000-0002-3083-436X

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
