## [Reviewer comments · BMJ Open]

ARTICLE DETAILS

TITLE (PROVISIONAL)	Predicting the risk of asthma attacks in children, adolescents and adults: protocol for a machine learning algorithm derived from a primary care-based retrospective cohort
AUTHORS	Hussain, Zain; Shah, Syed Ahmar; Mukherjee, Mome; Sheikh, Aziz

VERSION 1 – REVIEW

REVIEWER	Katsuyuki Tomita, M.D., Ph.D. Yonago Medical Center, Japan
REVIEW RETURNED	27-Dec-2019

GENERAL COMMENTS	I'm afraid to say that there is no "Result" section in this manuscript.
---

REVIEWER	Amanda Messinger MD University of Colorado Denver School of Medicine Children's Hospital Colorado
REVIEW RETURNED	31-Dec-2019

GENERAL COMMENTS	Exciting work. Well thought out methods and clear description of ML techniques. 1. Regarding determination of BTS step feature (in Table 1), will this determination be based on analysis of prescribed medications at a certain time, documentation in the note, or read codes?2. I would like more detail about your exclusion criteria; are you including patients with co-morbidities such as COPD or interstitial lung disease, or other serious respiratory ailments? I understand the desire to get a "real world cohort" but the concern for confounding is substantial.3. What (if any) is the target time window you are hoping to predict? 1 week, 2 months, 6 months before exacerbation?4. Does the data set include prescription fill data as a proxy for adherence or non-adherence? Any other features that might capture medication use?
---

REVIEWER	Seyyed Shamsadin Athari zanjan university of medical sciences
REVIEW RETURNED	30-Jan-2020

GENERAL COMMENTS	The manuscript with entitle "Predicting the risk of asthma attacks in adolescents and adults: protocol for a machine learning algorithm derived from a primary care-based retrospective cohort" is acceptable for publication in this journal but there are some points that would be better to correct before publication. In the children under 12 years old Asthma has high prevalence. Why age was 12-80? What about under 12 years old? The pattern of study, method and analyzing need to more explain.
--

	Please revise the abstract. Table 1, Figures and supplements should be revised. The figures are confusing, please revise the figs. It is better that this manuscript would have overview in prevention, diagnosis, treatment and management phase of asthma.
--	---

REVIEWER	Quan Do Mayo Clinic - Rochester - MN, USA
REVIEW RETURNED	01-Feb-2020

GENERAL COMMENTS	It is quite an ambitious undertaking, comprehensive, deploying multiple machine learning methods/algorithms to see if there are significant differences (in terms of utility) in the results precipitated by each. This paper is acceptable as a protocol paper. The authors correctly stated the problem, issues of the different machine learning method. The study protocol has been described clearly. Conditional probabilities vs. unconditional probabilities are a big issue when applying machine learning methods in medical studies. I'm glad to see a study which is carried out to compare the one-class classifier and two-class classifier in a medical study. As each of these two methods have both limitations and strengths, it would be a good idea to come up with a generalization method which can minimize the limitations and maximize the strengths of both methods. Here is a suggestion. If the data permits, I would also recommend the addition of another data point. Studies have recorded differences, perhaps related to immunological differences, in asthma prevalence by ethnicity (or race). If the data permits results, such extension would be at minimal cost. Another suggestion is gathering the environmental exposures data (maybe through Asthma Smartphone APPs). This is a solid issue of asthma. Good luck to the authors; I look forward to the publication of findings.
---

VERSION 1 – AUTHOR RESPONSE

Reviewer: 1

I'm afraid to say that there is no "Result" section in this manuscript.

Thank you for your comment. Since this is a protocol paper, we have not undertaken the work proposed and hence there are no results.

Reviewer: 2

Exciting work. Well thought out methods and clear description of ML techniques.

Thank you very much for your support and positive feedback.

1. Regarding determination of BTS step feature (in Table 1), will this determination be based on analysis of prescribed medications at a certain time, documentation in the note, or read codes?

These will be determined by analysis of prescribed medications (also coded in READ codes)

2. I would like more detail about your exclusion criteria; are you including patients with co-morbidities such as COPD or interstitial lung disease, or other serious respiratory ailments? I understand the desire to get a "real world cohort" but the concern for confounding is substantial.

Thank you for this very important point. We aim to include patients with various co-morbidities and will not attempt to exclude patients on the basis of certain comorbidities. We have clarified this in the manuscript (reproduced below):

"We will not attempt to exclude patients with co-morbidities. We have, however, included comorbidities (see Table 1) that will allow us to adjust for any potential confounders arising from comorbidities."

3. What (if any) is the target time window you are hoping to predict? 1 week, 2 months, 6 months before exacerbation?

We aim to predict asthma attacks over 3-, 6-, 12- and 24- month periods. This was mentioned in the introduction under "Research Aims" and we have reproduced this below:

"2. Systematically apply several machine learning algorithms (both one-class classifier and two-class classifiers) to predict the risk of asthma attacks, over 3-, 6-, 12- and 24-month outcome periods."

4. Does the data set include prescription fill data as a proxy for adherence or non-adherence? Any other features that might capture medication use

We only have access to the prescription records in the primary care database to rely on. Pharmacy records would have given us additional information to help us better estimate patient adherence. However, in this study, we acknowledge this as a limitation, and we have now included this in the discussion accordingly (reproduced below).

“Furthermore, we do not have access to pharmacy records for prescription data (which may help us better estimate patient adherence to medication prescription) and would therefore use prescription records to determine patient usage which may not always be the correct.”

Reviewer: 3

“The manuscript with entitle "Predicting the risk of asthma attacks in adolescents and adults: protocol for a machine learning algorithm derived from a primary care-based retrospective cohort" is acceptable for publication in this journal “

Thank you for your comment.

“In the children under 12 years old Asthma has high prevalence. Why age was 12-80? What about under 12 years old?”

The comparable work (by Blakey et al. 2016) that we originally aimed to improve on in terms of developing prognostic models included patients aged 12-80. However, this is quite important to also look at children under 12 years of age. We have consequently modified our inclusion criteria to include children aged 8-12 years as well. We chose the cut-off of 8 years as this is the cut-off chosen by NICE as part of the Quality and Outcomes Framework for asthma diagnosis (<https://www.nice.org.uk/standards-and-indicators/qofindicators>). A potential predictor that might be useful to identify asthma attacks is eosinophil count. We have consequently modified the inclusion criteria and also included the eosinophil count as an additional potential feature in Table 1. The relevant changes are reproduced below:

“Identify significant risk factors associated with asthma attacks in children, adolescents and

adults (aged 8-80 years), and appropriately select these for inclusion in our analysis.

Eosinophil Count	Blood eosinophil count (cells/L) categorised into high and not high (threshold of 0.35×10^9 cells/L to define high/not high eosinophil count ¹³)
------------------	---

The pattern of study, method and analyzing need to more explain.

Please revise the abstract.

Table 1, Figures and supplements should be revised. The figures are confusing, please revise the figs.

We have modified the text in light of the other reviewer comments and we hope that this also addresses the concerns raised by Reviewer 3. However, if there are specific issues related to what exactly needs revision in abstract, table 1 or figures or supplement and what are the issues with them at the moment, then please let us know and we will revise accordingly.

Reviewer: 4

It is quite an ambitious undertaking, comprehensive, deploying multiple machine learning methods/algorithms to see if there are significant differences (in terms of utility) in the results precipitated by each. This paper is acceptable as a protocol paper. The authors correctly stated the problem, issues of the different machine learning method. The study protocol has been described clearly. Conditional probabilities vs. unconditional probabilities are a big issue when applying machine learning methods in medical studies. I'm glad to see a study which is carried out to compare the one-class classifier and two-class classifier in a medical study. As each of these two methods have both limitations and strengths, it would be a good idea to come up with a generalization method which can minimize the limitations and maximize the strengths of both methods.

Thank you very much for your support and positive feedback.

If the data permits, I would also recommend the addition of another data point. Studies have recorded differences, perhaps related to immunological differences, in asthma prevalence by ethnicity (or race). If the data permits results, such extension would be at minimal cost.

This is very helpful and we would aim to get this information and incorporate in our model. At the moment though, we do not have access to ethnicity information (but the OPCRD does contain linkage data for both ethnicity and deprivation data that we will aim to acquire during the course of this project). We have amended Table 1 accordingly and added a new row about ethnicity as a candidate predictor (reproduced below).

Ethnicity	Ethnicity information if available (White, Black, Asian, South Asian Caribbean etc.)
-----------	--

Another suggestion is gathering the environmental exposures data (maybe through Asthma Smartphone APPs). This is a solid issue of asthma.

Thank you for the suggestion and environmental exposure data is indeed going to be very relevant for asthma attack prediction. Unfortunately, the dataset we have is anonymised and as a condition of use, we will make no attempt to identify/contact the patients we analyse (and we currently do not have the capacity to collect such information at national level).

VERSION 2 – REVIEW

REVIEWER	Quan Do Mayo Clinic, USA
REVIEW RETURNED	01-Apr-2020
GENERAL COMMENTS	The paper is in the editing format with all comments?